# LLM DETECTORS STILL FALL SHORT OF REAL WORLD: CASE OF LLM-GENERATED SHORT NEWS POSTS

## ABSTRACT

With the emergence of widely available powerful LLMs, disinformation generated by large Language Models (LLMs) has become a major concern. Historically, LLM detectors have been touted as a solution, but their effectiveness in the real world is still to be proven. In this paper, we attempt such an evaluation on news-like posts generated by moderately sophisticated attackers, critical in information operations. We develop a new LLM testing methodology, accounting for the generative LLMs and detector LLMs topic performance heterogeneity, and accounting for the inherently adversarial setting, and demonstrate that under our threat mode, SotA detectors can be evaded with a trivial temperature attack, which could not be predicted from prior large-scale benchmarks, that not only failed to test for it but also suggested significantly better performance than what we observed in closely matching settings. Finally, we demonstrate that LLM detection in the real world involves a tradeoff between the generalization to unseen adversarial evasion attacks and the generalization to unseen variations of human texts, which have not been tested until now.

We believe our findings suggest that current large-scale LLM benchmarking practices are not relevant to the real-world setting and offer an alternative method, deriving a dynamic benchmark from a set of human texts based on a threat model, which we publicly release at `https://anonymous.4open.science/r/benchmark_ai_news_detection-873E/README.md`.

## 1 INTRODUCTION

The misuse of large language models (LLMs) for propaganda, inciting extremism, or spreading disinformation and misinformation has been a major concern from the early days of LLMs development (Bender et al., 2021; McGuffie & Newhouse, 2020; Ippolito et al., 2020). While this concern has led LLM developers to withhold their most powerful models in the past (Solaiman et al., 2019), powerful LLMs and multimodal generative models have now been published despite such concerns persisting (OpenAI, 2023). With the release of powerful open-weight LLM families that can be deployed on commodity hardware such as LLaMA (Touvron et al., 2023), Phi (Gunasekar et al., 2023) or Zephyr (Tunstall et al., 2023), LLM misuse can no longer be mitigated through model updates or inputs/outputs monitoring.

### 1.1 LLMs IN INFORMATION OPERATIONS

A common goal of information operations (intentional and coordinated efforts to modify the public perception of reality with ulterior motives) is to modify the perception of the current situation (Goldstein et al., 2023; Maschmeyer, 2022). A well-documented and effective approach to achieve that uses social networks by seeding them with news-like narratives. While the original seeding might have a minimal impact, some narratives will trigger a strong partisan response and will be re-shared by key genuine users, who will customize the narrative for their audience and engage in debate to further it (Eady et al., 2023; Baribi-Bartov et al., 2024). While in some cases such misinformation is picked up by traditional news outlets, adding to its credibility, the quantity is a quality of itself, and continuous repetition of the same narratives through different channels leads to their acceptance as

truths (DiRESTA, 2018; Geissler et al., 2022). Given the competition for attention on social media, both seeded narratives and legitimate news tend to be shared as short-form pots of 260-520 characters with links and images. Such format is common to Twitter (now X), Bluesky, Meta Threads, and the largest Mastodon instance.

In the past, this technique has been shown to be effective in spreading counterfactual claims and hard to mitigate (Vosoughi et al., 2018). However, large-scale information operations of this type have been easily detectable until recently, due to exact text reuse, inconsistencies in online persona, and lack of non-trivial interaction with other users. Such failures are understandable, given the scope of the activities and the cost of human operators. However, LLMs drastically alter the cost-quality tradeoff (Musser, 2023). Combined with the reports that LLMs are better than humans at both personalized political persuasion and disinformation concealment (Matz et al., 2024; Li et al., 2024; Chen & Shu, 2023), LLM-augmented information operations promise to be radically more effective and more difficult to detect and counter, posing a serious immediate threat. As such, it is a prime real-world domain of application for LLM detectors.

## 1.2 LLM DETECTABILITY

Unfortunately, humans do not distinguish well LLM-generated texts from human-written ones (Ippolito et al., 2020; Matz et al., 2024; Jakesch et al., 2022). Early in the generative LLM development (Zellers et al., 2019; Solaiman et al., 2019) proposed that accurate LLM detectors could mitigate that issue. Unfortunately, follow-up research rapidly discovered that LLM detectors failed if generation parameters were altered (Ippolito et al., 2020; Fishchuk & Braun, 2023), output was paraphrased (Krishna et al., 2023a), or barely more complex prompts were used (Wu & Hooi, 2023a; Bakhtin et al., 2019).

The issue gained in salience with the release of ChatGPT, leading to several large-scale LLM detector benchmarks (Li et al., 2024; Sadasivan et al., 2024; He et al., 2023; Wu & Hooi, 2023a), and most recently Dugan et al. (2024); Wang et al. (2024a). Unfortunately, with the disparate performance of detectors for different types of texts, most benchmarks fail to account for the inherently adversarial setting of LLM detection, and suffer from the same pitfalls as many other ML-based security solutions (Arp et al., 2022). Hence, there is still no consensus as to whether LLM detectors are ready for real-world applications (Da Silva Gameiro, 2024). Here, we try to address both the methodological issues and decide on the detectors' real-world usefulness in our setting.

## 1.3 SETTING AND ATTACKER CAPABILITIES

An attacker with arbitrary capabilities is neither realistic, nor can realistically defended against. Consistently with subsection 1.1, we focus on moderately sophisticated attackers. We assume they are capable of deploying on-premises SotA LLMs in the 1-10B parameter range and familiar with adversarial evasion strategies available at inference, such as generation parameters modification (Fishchuk & Braun, 2023; Ippolito et al., 2020), paraphrasing (Krishna et al., 2023a), or alternative prompting strategies (Wu & Hooi, 2023a; Bakhtin et al., 2019). We assume that the attacker cannot evasively fine-tune the generative models.

We assume that the attacker is seeking to generate news-like content of approximately 500 characters, as a completion of a headline or opening sentence, that needs to be detected by a social media operator in an environment where LLM-generated news-like content is not predominant and true positive labels are not available, requiring a low target false positive rate (FPR), which we set to 5%. Finally, we assume that providers of LLM models over API perform information operation detection and suppression.

## 1.4 CONTRIBUTIONS

- We formulate the LLM detection task as inherently adversarial in the real world and consolidate best practices for defensive tool evaluation in that setting
- We integrated these practices into a dynamic, extensible benchmarking tool to generate a testing dataset given reference domain texts and a threat model

- We show that SotA zero-shot detectors are vulnerable to trivial adversarial evasion in our setting, which is not reflected by prior benchmarks.

- We show that developing LLM detectors involves a tradeoff in generalization to evasion attacks and variability of human writing, previously unreported and untested

- We comprehensively benchmarked custom detector training strategies and demonstrated that while they can be used to develop a well-generalizing detector, it would not be ready for real-world use

Based on these results, we conclude that LLM detectors are currently not ready for countering LLM-aided information operations and that the current approach to LLM testing with large-scale static benchmarks is unsuitable for real-world relevance.

## 2 BACKGROUND AND RELATED WORK

### 2.1 TRAINED DETECTORS

LLMs have led to a pardigm shift, from purpose-training ML models, to fine-tuning base models. Rather than re-training a new model for each application from scratch, a large model is first pretrained on a large quantity of data, and is then fine-tuned to adapt it to downstream task. This paradigm is particularly well-suited for classification tasks as demonstrated by (Devlin et al., 2019). As we can formulate detection as a classification task, multiple papers take this approach, historically pioneered by (Solaiman et al., 2019). It is also currently considered a SotA approach for training custom LLM detectors (Ippolito et al., 2020; He et al., 2023)

**BERT-based detectors**  BERT (Devlin et al., 2019)) and its subsequent improved versions, RoBERTa and Electra (Liu et al., 2019; Clark et al., 2020)), have gained widespread adoption for classification tasks. Their bi-directional encoding architecture offers an advantage over the autoregressive decoder-only architecture, given the ability to account for the context of both the preceding and following context rather than just the preceding, as well as obligatory anchoring to the provided text. Moreover, their relatively small size, such as the 300M parameters for RoBERTa Large compared to the 175B parameters for GPT-3, makes them highly practical.

### 2.2 ZERO-SHOT DETECTORS

Trained detectors show promising results in multiple works such as in Mitrović et al. (2023). However, as highlighted by Wang et al. (2024b), these trained detectors fail to generalize to text distribution shifts. Zero-shot detectors such as in Mitchell et al. (2023) and Bao et al. (2024) can be seen as an appealing alternative to mitigate this issue. Zero-shot detectors, i.e., detectors not relying on training, are more suitable when we try to detect text not coming from a specific distribution. They are also easier to use in practice since we do not need to train the detector on the domains, although at the price of generally being more resource-intensive due to using larger models.

**Fast-DetectGPT**   is a SotA open-source detector, reported to perform well across domains and common attack strategies (Bao et al., 2024), scoring as an overall top performer for FPRs < 5% in adversarial third-party benchmarks (Dugan et al., 2024).

**GPTZero**   is one of the most widely used commercial LLM detectors (Tian & Cui, 2024), consistently included into third-party benchmarks and similarly a top performer (Dugan et al., 2024).

**RoBERTa-Base-OpenAI**   is a BERT-based detector fine-tuned by OpenAI to detect GPT2 output (Solaiman et al., 2019). While it is not a zero-shot detector per se, it is still commonly benchmarked and used as such, with some benchmarks reporting good performance (Wang et al., 2024a). At the time of writing (May 2024), it was downloaded over 159,000 times in the previous three months, according to its HuggingFace repository.

## 2.3 BENCHMARKING DETECTORS

**Adversarial evasion** setting is inherent to LLM detectors; its performance cannot be detached from performance against potential attacks. Multiple works have attempted to tackle this issue, generally focusing on specific attacks. For instance, Krishna et al. (2023b) and Wu & Hooi (2023b) introduced a paraphrasing attack they demonstrated to be efficient, while Bao et al. (2024) investigated evasion through alternative decoding strategies.

**Generative generalization** benchmarking, be it across LLMs (He et al., 2023; Pu et al., 2023) or domains (Li et al., 2024; Xu et al., 2023) is equally essential. LLM generation can be useful to an attacker in multiple contexts, and with a proliferation of widely available powerful LLMs, an attacker cannot be assumed to restrict themselves to a single generative model.

**Human-text generalization** evaluation is, unfortunately, all but absent from LLM detectors benchmarks, despite reported issues in real-world usage, e.g. by Liang et al. (2023b). Existing benchmarks report - at best detection rates for additional domains, without reporting FPRs.

Unfortunately, systematic studies remain few, and the recent benchmarks attempting them, such as Dugan et al. (2024); Wang et al. (2024a) suffer from issues, notably failing to include common attacks, such as temperature increase, first described by Ippolito et al. (2020), and evaluation of FPRs on diverse human texts. As static benchmarks optimized for detector evaluation, they are difficult to add new attacks to or transfer to new domains, making them unsuitable for our application.

## 3 METHODOLOGY

All results can be replicated with code provided in the benchmark repository: `https://anonymous.4open.science/r/text_llm_detector-3E07`, to be published under the MIT license. English is the only language considered here, with all datasets, prompts, and fine-tuning data in English.

### 3.1 DETECTOR MODELS

To evaluate a domain-specific pretrained detector, generally reported to be one of the best detection methods (Wang et al., 2024a; He et al., 2023), we evaluated three base pre-trained LLMs: RoBERTa-Large, Distil-RoBERTa, and Electra-Large[1]. RoBERTa and Electra have been shown to achieve improved results over BERT on fine-tuning to classification tasks in Liu et al. (2019) and Clark et al. (2020) thanks to a different pre-training method (Liu et al., 2019; Clark et al., 2020). We added Distil-RoBERTa to evaluate the performance of a smaller model (82.8M parameters), more usable at scale.

To test zero-shot detectors, we evaluated the three detectors mentioned previously: RoBERTa-Base-OpenAI, due to its usage, and Fast-DetectGPT and GPTZero, generally considered as representatitve of SotA open-source and commercial detectors, respectively (Dugan et al., 2024).

### 3.2 GENERATING THE DATASETS

We chose six different generator LLMs. Three non-chat foundational models, Phi-2, Gemma-2B, and Mistral-0.1, are the only ones used to train the custom detectors. Three chat models, Gemma-chat, Zephyr, and LLama-3-8B-Instruct, are only used for testing and are representative of commonly used SotA open-weight LLMs.

To create neural fake news, leveraged the CNN Dailymail news dataset, representative of American English news. To obtain the LLM-generated articles, we take a news article from the dataset, clean the beginning of the article to remove header content, and then pick the 10 first words of the article as a prefix. We use this prefix as the prompt for the non-chat models to generate the rest of the article and prefixed it with a supporting prompt for chat models (see appendix B.2). We let the model generate up to 200 tokens but cut the generation to have only the first 500 characters. We also cut the original articles to 500 characters to obtain the reference human samples. Using this procedure, human and generated samples are indistinguishable in length (see examples in appendix B.3).

---

[1]All download links for models, datasets, and code are available in the appendix table 7, in the appendix H

By using the method described above, we create 6 datasets with around 20K samples each (see appendix B.1 for the precise sizes) with a train, eval (10% of the whole size), and test split (10% of the whole size). In addition to the above procedure, we pair the AI-generated samples with their corresponding original news articles. We do that to prevent the fake and corresponding true samples from being in different training batches or data splits. We also filter out the generated news articles under 500 characters and discard the corresponding true sample to keep the dataset balanced. Finally, we also create a "round robin" dataset, which consists of a mixture of data generated by the 3 non-chat models, expected to train models more suitable for adversarial setting (Kucharavy et al., 2020).

## 3.3  TRAINING THE DETECTORS

We finetuned the pretrained models described in subsection 3.1 on each non-chat model training dataset and the round-robin one. The detectors are trained on full datasets for 1 epoch with an evaluation after every 200 samples. We save only the model that obtained the best loss on the evaluation set to avoid overfitting. We list the hyperparameters for each training procedure in appendix C.1.

While we tested different ways of finetuning the detectors on the datasets, we only retained full finetuning for the results section, consistently with the detector training SotA recommendations. We also provide some results when only finetuning a classification head and using the adapters PEFT method in appendix E. We tracked the model performance on the pretraining task to confirm we were not overfitting the base models to our distribution (cf appendix F).

## 3.4  TESTING THE DETECTORS

We use the same metric across all the results to test the detectors: TPR (True-Positive rate). To obtain the detection prediction, we use a threshold on the output of the detector to target an FPR (False-Positive rate) of at most 5%. We find these thresholds by finding the TPR/FPR at different thresholds on the evaluation set for each dataset. We repeat this threshold-finding procedure for each detector (trained and zero-shot). This setting mimics a realistic scenario where a defender uses a detector to target a low level of false positives without access to true positive labels. We do not recompute a detection threshold on each attack since we consider these attacks unknown to the defender.

### 3.4.1  TESTING FAST DETECT GPT

To test Fast-DetectGPT (Bao et al., 2024), we use the script from their GitHub repository that we slightly adapted to run similarly to the other detectors we tested. We use GPT-J 6B B as the reference model since it provides more accurate results according to their work.

### 3.4.2  TESTING GPTZERO

To test GPTZero (Tian & Cui, 2024), we used an academic partner API access, courtesy of GPTzero, version `base-2024-04-04`. In order to maintain consistency with the methodology presented here, we forced the FPR of GPTZero to 5%, given that by default, it is heavily biased to minimize FPR ($< 0.3\%$ in our setting).

## 3.5  ATTACKS AGAINST THE DETECTORS

### 3.5.1  EVASION ATTACKS

In experiments targeting detector evasion, we start with the test set created in the previous experiment when generating fake articles. We regenerate the fake news articles using a different generation parameter or a different prompt, obtaining a new dataset sharing the same true original articles as before, but differing from the one used to produce the training and test data previously. Consistently with prior research Sadasivan et al. (2023), we assume the evasion attacks do not alter the text enough to make it unusable.

**Changing the Generation Parameters**   The generation parameters we modify are the temperature and the repetition penalty. The first attack, called "high temperature," sets the temperature to 1.2 (note that OpenAI's API allows a temperature up to 2). The second, which we call "repetition penalty," consists in setting the repetition penalty to 1.2 (interestingly, hugging chat uses a repetition penalty of 1.2). Both parameters heavily impact the diversity[2] of the produced text, making it more difficult for detection methods that rely on the lack of diversity of AI-generated texts.

**Prompting Attacks**   We consider a few prompts to generate the fake news articles (the complete list can be found in appendix B.4). The prompts we choose should cover a wide set of prompts an attacker may use. The idea for our experiment is to consider only basic attacks, i.e., attacks that do not require training an additional LLM (low resource) or special LLM expertise (low skill). While we are aware that more complex attacks exist, such as those described in the background (see Huang et al. (2024) for example, which uses multiple LLMs to generate text), our intent is not to test exhaustively all possible attacks (which is not feasible), but to limit ourselves to the setting of our threat model. Our results show that we do not need to dig too far to find effective attacks, which is even more concerning.

**Paraphrasing Attacks**   Paraphrasing a generated text with an auxiliary LLM has been reported to achieve good evasion against a variety of detectors (Wu & Hooi, 2023b). Given that it preserves semantic meaning while modifying probabilities of token selection, it is likely to erase non-semantic traces of LLM generation and, as such, to be highly performant. Here, we implement a simpler version of the attack than DIPPER (Krishna et al., 2023b), prompting an LLM to reformulate the text (cf Figure 6 )

### 3.5.2   TESTING THE DETECTORS ON HUMAN TEXT FROM A DIFFERENT DISTRIBUTION

Finally, we also test the detectors on 10000 samples from the Xsum dataset. Xsum comprises articles from BBC News, a different news outlet from the one we used to until now, based on CNN. XSum notably uses British English. This experiment tests the human-text generalization of detectors, so we did not create fake news articles.

## 4   RESULTS AND DISCUSSION

### 4.1   TRAINED DETECTORS

Here, we present TPRs at FPR fixed to 5% for detectors where we finetune all parameters. Results obtained with other finetuning methods can be found in the appendix E.

**Performance on training LLM**   We present in figure 1 the TPR obtained when testing the trained detectors on the same dataset they have been trained on.

In our experiments, Electra (Electra-Large) consistently outperforms or comes equal with the other detectors. We can also notice that the difference is small between the distilled RoBERTa version (82.8M parameters) and its large version (355M params parameters). For the next results, we will show only the results using Electra as the detector, but the results for other detectors are available in the appendix D. We also note that TPR with fixed FPR provides a more fine-grained performance evaluation compared to all-around stellar ROC-AUC values in Appendix Figure 9 and that adapter and classification head underperform compared to full finetuning (cf. appendix E).

**Generalization across training LLMs**   We show on the heatmap in figure 2 the TPR of the trained detectors on the test of the different datasets generated by the LLMs, evaluating finetuned detector generalization across generating LLMs.

Surprisingly, most detectors perform almost equally, if not better in some cases, when detecting fake news articles generated by a different LLM than the one used to train them. This suggests a good

---

[2]By diversity, we mean the entropy of the logits for the next token prediction. Higher entropy means that the range of the next word that can be generated is higher, resulting in higher text diversity.

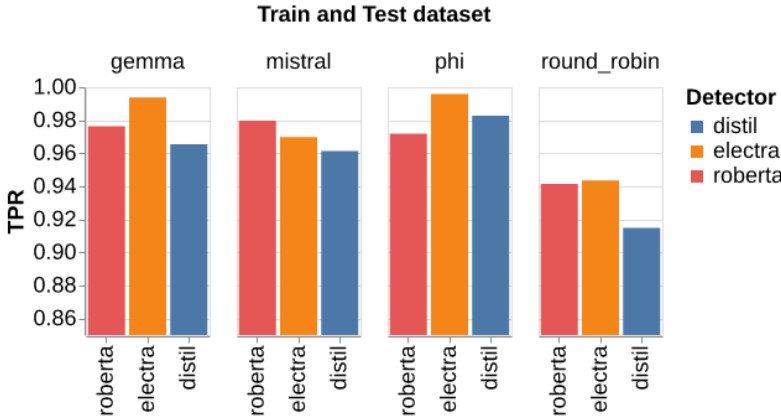

Figure 1: Detector TPR when testing them on the test set of dataset they have been trained on. "round_robin represents" a mixture of all the other datasets. Roberta is RoBERTa-Large, Electra - Electra-Large, and distil - Distil-RoBERTa-base. The TPR is for a target FPR of at most 5%.

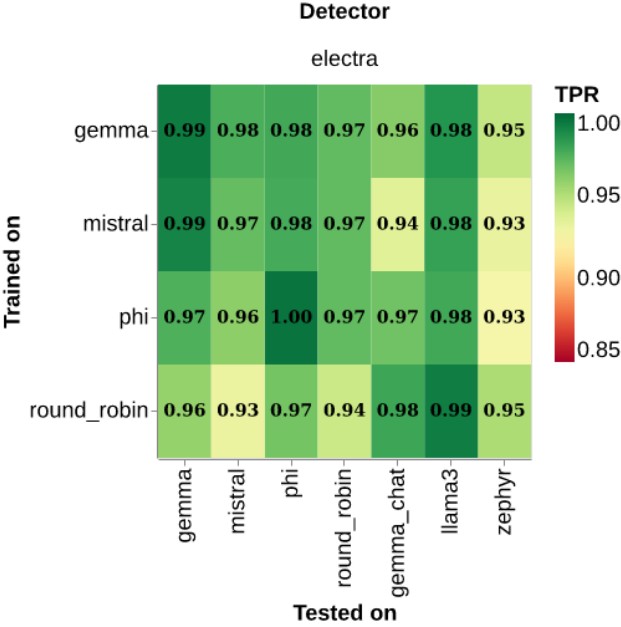

Figure 2: Detector TPR for the different Electra-Large finetuned models when testing them with datasets generated by different LLMs. Gemma is Gemma-2B, Phi is Phi-2, and Round-Robin is a mixture of gemma, mistral, and phi. Gemma_chat is gemma_2B-it, Llama3 is Llama3-Instruct-8B, and Zephyr is Zephyr-7B-Beta. The TPR is for a target FPR of at most 5%.

generalization of the patterns leveraged by the finetuned detectors and a sufficient similarity in LLMs output in our setting.

**Generalization to unseen chat LLMs** In this experiment, we repeat the same testing as above, but on 3 datasets generated by chat models unseen at training (see appendix B.2 for the prompt). The results are in the same heatmap as for the previous experiment on figure 2.

We find that the TPRs obtained here align with the finding above that the trained detectors surprisingly generalize enough to achieve similar TPR across data generated by instruction-tuned LLMs. We find that news articles generated by Zephyr are slightly harder to detect, whereas the ones generated by Llama 3-Instruct-8B are the easiest. This highlights that while we find encouraging generalization

results across LLMs, the performance across LLMs remains disparate, consistently with prior findings (Pu et al., 2023).

## 4.2 CROSS-LLM GENERALIZATION

We present in figure 3 the results of the same experiment as above, but including zero-shot detectors, which we compare to one of the trained detectors. We observe that Fast-DetectGPT and GPTZero closely match but still underperform compared to Electra-Large trained to detect Mistral-generated texts (Electra_Mistral). We also observe that RoBERTa-OpenAI significantly underperforms, suggesting that it is now outdated and strongly arguing for abandoning it as a zero-shot general LLM detector.

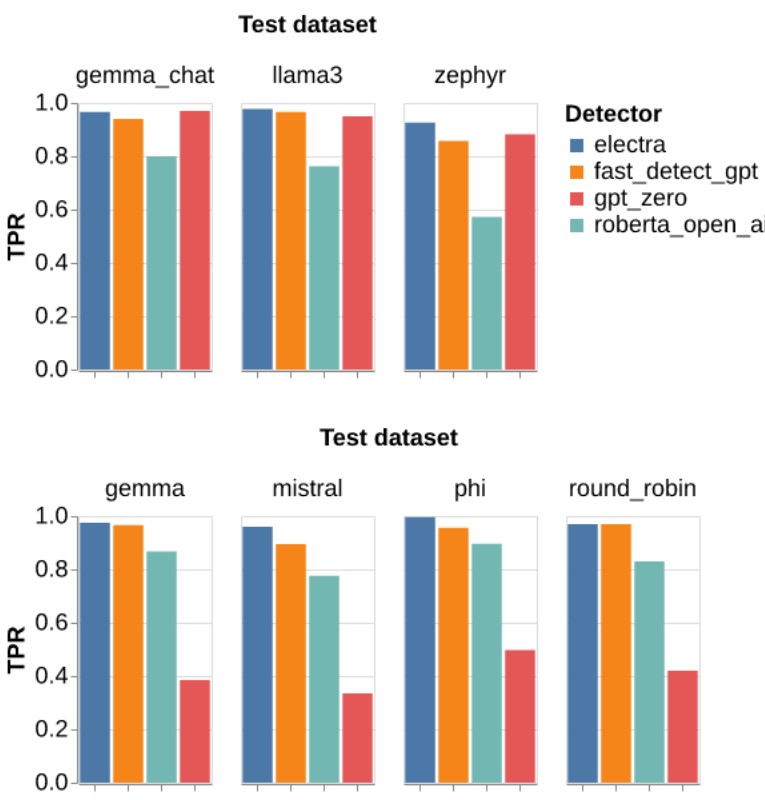

Figure 3: TPR comparison of detectors tested on the dataset generated by the chat models (upper part) and non-chat models (bottom part). Electra is Electra-Large finetuned on Mistral samples, and roberta_open_ai is the RoBERTa detector released by OpenAI. The TPR is for a target FPR of at most 5%.

Again, we find Zephyr LLM particularly challenging for all detectors and, going forward, will focus on it to best differentiate model performance. Curiously, we also observe that GPTZero drastically under-performs on non-chat models, consistently with previous benchmarks (Dugan et al., 2024), suggesting an overfitting to chat models.

## 4.3 EVASION ATTACK RESLIENCE

Here, we focus on the best-performing trained detector (Electra_Mistral) and the most challenging generator - Zephyr. We also include a detector trained on the mixture of data from all generators in a round-robin fashion (Electra_RR), which we expect to generalize well. We also include the previously mentioned zero-shot detectors. Following the method described in the Methodology (3.5), we generate evasion attack datasets and present TPR (true-positive rate) at a fixed FPR of 5% in the table 1, as described in 3.4.

| Detector | No Attack | High temperature | Repetition penalty | Attack
News prompt | Tweet prompt | Example prompt | Paraphrasing |
|---|---|---|---|---|---|---|---|
| Electra_Mistral | 93% ± 1% | ↓16% 77% ± 2% | ↓34% 59% ± 2% | ↓3% 90% ± 1% | ↓35% 58% ± 2% | ↓3% 90% ± 1% | ↓22% 71% ± 2% |
| Electra_RR | **95%** ± 1% | ↓7% **88%** ± 1% | ↓10% 85% ± 2% | ↑1% 96% ± 1% | ↓11% 84% ± 2% | ↓2% **93%** ± 1% | ↓1% **94%** ± 1% |
| RoBERTa-OpenAI | 57% ± 2% | ↓14% 43% ± 2% | ↓31% 26% ± 2% | ↑5% 62% ± 2% | ↓33% 24% ± 2% | ↓3% 54% ± 2% | 0% 57% ± 2% |
| Fast-DetectGPT | 86% ± 1% | ↓32% 54% ± 2% | ↓62% 24% ± 2% | ↓2% 84% ± 2% | ↓44% 42% ± 2% | ↓3% 83% ± 2% | ↓41% 45% ± 2% |
| GPT-Zero | 88% ± 1% | ↓34% 54% ± 2% | ↑1% **89%** ± 1% | ↓3% 85% ± 2% | ↑11% **99%** ± 2% | ↓28% 70% ± 2% | ↑2% 90% ± 1% |

Table 1: TPR obtained when testing the detectors on the adversarial generated test set with the threshold to target an FPR of at most 5%. Electra_Mistral is Electra-Large finetuned on Mistral samples, and Electra_RR is Electra-Large finetuned on the Round-Robin dataset we created. All adversarial texts here are generated using Zephyr (see table 6 for results when detecting adversarial texts generated with other LLMs using the same evasion attacks).

**Changing the Generation Parameters** We observe that both attacks against generation parameters are highly effective, with a modest temperature increase defeating all zero-shot detectors. This is highly concerning, given that this attack is trivial well-known (Ippolito et al., 2020; Solaiman et al., 2019). This is an example of attacker using benchmarks to find attacks the defender is not expecting.

We hypothesize that by increasing the temperature and repetition penalty, we increase the LLM output diversity, which defeats LLM detectors based on text perplexity and expecting generated texts to be less diverse than human-written ones. We hypothesize that the poor performance of detectors against the temperature attack is due to its absence from any recent LLM detector benchmarks.

**Prompting the LLM** For the prompting attacks, we discover a high variation in terms of both the average effectiveness of the attacks and model-specific attack effectiveness, with none achieving universal effectiveness. The "News prompt" attack seems to be largely ineffective, while the "Example prompt" is only effective against GPTZero, and the "Tweet Prompt" attack is effective against all detectors but GPTZero, whose performance instead drastically increases.

We hypothesize that this variation is a result of a combination of factors, including the selection of reference human/machine datasets by detector developers and the domain-specific proficiency of generators. We believe this highlights the critical need for diverse testing of detectors, both domain-specific and threat-specific, and for exploring a wide range of prompting strategies.

**Paraphrasing** We observed that the paraphrasing attack was only effective against Electra_Mistral and Fast-DetectGP, with the latter losing 40% TPR. This is somewhat unexpected, given prior reports of the effectiveness of this attack, although potentially due to using a prompted LLM reformulation rather than the DIPPER attack (Krishna et al., 2023b).

**Prior Benchmark comparison** A direct comparison to prior benchmarks is non-trivial. Given the drastic difference in performance of different prompting strategies, minor differences in benchmark implementation matter. In our case, the only benchmark providing sufficiently detailed results is RAID Dugan et al. (2024). In its leaderboard, we would expect the results for their "News" domain (BBC articles), generated by the Mistral-chat model, to correspond to our setting of news-like content (CNN articles trimmed to 500 characters), generated by the Mistral-based Zephyr.

Unfortunately, this is not the case. Even without attacks, RAID suggests RoBERTa-Base-OpenAI-GPT2 matches FastDetectGPT (0.99 TPR for both), whereas we observe a drastically lower and more heterogeneous performance (0.57 and 0.86 TPR respectively). A repetition penalty attack, sharing the same parameters for both our and RAID benchmarks leads to a similar difference (0.5 and 0.4 TPR in RAID, respectively, vs. 0.26 and 0.24 TPR for us). Likewise, for the paraphrasing attacks, RAID predicts Fast-DetectGPT to remain highly effective (0.95 TPR), while we observe its performance halved (0.45 TPR), which is even more surprising given that we used a weaker paraphrasing attack.

This means that domain-specific and threat model-specific benchmarking is essential to judge the real-world performance of LLM detectors. In our opinion, this argues against the utility of large-scale benchmarks with once-generated static data and in favor of dynamic, application-specific benchmarks.

## 4.4 Performance on Unseen Human Texts

In the previous subsection, we showed that while a moderately sophisticated attacker could defeat all detectors, one of our custom-trained detectors - Electra_RR - exhibited an outstanding resilience to evasion attacks with a usable and consistent 0.84 TPR. Such resilience is surprising, given it was not trained against adversarial evasion and did not generalize particularly well across generative LLMs, and could lead us to suppose that Electra_RR would perform well in new evasion attacks, hence claiming a new SotA.

However, such a claim would be misplaced. In real-world deployments, LLM detectors must not only generalize across unseen generators and attacks but also across unseen human texts, maintaining an acceptable FPR across different types of texts. We test for this in table 2, verifying the generalization from the CNN News dataset to the BBC News Xsum dataset, which are closely related and differ most likely only by US English vs British English usage. Despite such close relatedness, we observe that Electra_RR fails dramatically in human text generalization along with Electra_Mistral, indicating that neither of our trained detectors can be deployed to the real world.

| Detector | Dataset | |
| --- | --- | --- |
| | CNN (only human) | BBC (only human) |
| Electra_Mistral | 96% ± 1% | ↓28% 68% ± 1% |
| Electra_RR | 94% ± 1% | ↓34% 60% ± 0.5% |
| RoBERTa OpenAI | 81% ± 1% | ↓5% 76% ± 1% |
| Fast-DetectGPT | 95% ± 1% | 0% 95% ± 0.2% |
| GPTZero[3] | 93% ± 1% | ↓3% 90% ± 0.3% |

Table 2: Accuracy of detectors on fully human-written datasets. All variation is due to false positives. For all detectors, FPR was set as close to possible to 5% on the known human set (CNN)

Unfortunately, such tests for out-of-distribution human texts are all but absent for LLM detector benchmarks, with only anecdotic reports of real-world performance failure (Liang et al., 2023a) raising awareness of this failure mode. Unlike what we could expect from prior works showing generalization of LLM detector across languages (Macko et al., 2023), and an effective underlying pretrained model to our classifier, the human text generalization cannot be assumed and must be tested in any LLM detector benchmarks for them to be useful.

## 5 Conclusion

In this paper, we developed a rigorous framework to benchmark LLM detectors in a way specific to a domain and threat model. By applying it to a setting relevant to LLM-augmented information operations, we show that current LLM detectors are not ready for real-world use due to a combination of susceptibility to trivial evasion attacks, notably generation temperature increase, and potentially unacceptable FPRs in practice, consistently with noted issues in other domains (Arp et al., 2022). Overall, we believe our results argue in favor of alternative approaches in that context, e.g., coordinated activity patterns search (Pacheco et al., 2021).

In the process, we noted several methodological issues with existing LLM detector benchmarks, notably missing evaluation of FPR on out-of-distribution human texts, incomplete coverage of evasion attacks, and the failure of large-scale benchmarks to predict detector performance in even closely related settings. We believe this argues in favor of dynamic benchmarks that can be run against a specific domain and threat model. To allow that, we release the dynamic benchmarking suite we developed (`https://anonymous.4open.science/r/benchmark_ai_news_detection-873E/README.md`), designed to be fully domain and language transferable for non-expert teams, and expandable with new attacks for more technical teams.

---

[3] Please note that GPTZero FPR has been forced to 5% here. With default configurations, the accuracy is over 99.7% on both datasets.

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

# A APPENDIX

## LIMITATIONS

While we focus on a setting highly relevant to in-the-wild LLM text recognition, our work has several limitations.

First, we focus on a short-story setting and use a well-known NLP dataset as a reference human text. Due to NLP dataset reuse for model training, the performance of the generative task is likely to be higher than that of actual information operations. Similarly, social media accounts without posting history are rare, and puppet accounts are likely to have similarly LLM-generated past posts. Such longer texts could increase confidence that an account is unauthentic. However, misappropriation of social media accounts for information operations is common, and such unauthentic post history is not guaranteed.

Second, we have not investigated text watermarking approaches. If future on-device LLM releases more tightly integrated models and supporting code and enables watermarking, it could potentially improve the detectability of such posts. While not impossible, recent work by Sadasivan et al. (2024) suggests that reformulation attacks remain effective in this setting, warranting caution as to their effectiveness.

Third, we only tested a limited number of zero-shot detectors. While it is possible that untested detectors could clear both adversarial evasion and generalization tests, it is unlikely. Bao et al. (2024) and Dugan et al. (2024) performed extensive benchmarking of recent zero-shot detectors, and both Fast-DetectGPT and GPTZero closely match other solutions across a large palette of tests, including adversarial perturbations.

Fourth, we assume the attacker has limited capabilities, notably lacking adversarial evasive model fine-tuning. This assumption is somewhat brittle, given that parameter-efficient fine-tuning (PEFT) requires minimal resource overhead compared to traditional fine-tuning. PEFT can be performed on quantized models from relatively small datasets and hardware adapted for quantized inference, putting them within the reach of moderately sophisticated attackers we consider. While this opens a new type of attack, current LLM detectors are easy enough to fool even without it.

Finally, our work focused on the English language, for which extensive resources are available and are leveraged by LLM developers. While our results could generalize to other high-resource Romance Languages such as French or Spanish, LLM performance in other languages, especially low-resource ones, is unlikely to induce native speakers into confusion, making the LLM disinformation detection a significantly less salient problem as of now.

## ETHICS STATEMENT

While the issue of deep neural disinformation is critical and a central concern for malicious misuse of LLMs, it has not prevented the release of powerful SotA models to the general public that can readily assist in such operations. Here, we do not present any novel attacks but demonstrate that existent ones are sufficient to evade SotA attacks. As such, we do not expect novel risks to arise from this work but rather to improve the general awareness of common tools' limitations and contribute to mitigating known risks arising from LLMs.

GPTZero is a security solution numerous entities use to detect LLM-generated text in potentially safety-critical contexts. Given that in this work, we found several highly effective attacks against it, we performed a coordinated vulnerability disclosure with them.

We used 1 A100 GPU on a local cluster to generate the datasets, 30 minutes per dataset. We used 1 A100 GPU to train the models and perform detection using Fast-DetectGPT. For testing, we only used V100 GPUs thanks to our focus on smaller models. In total, we used the local cluster for 30 hours of GPU-days on the A100 and 20 hours of GPU-days on the V100 (numbers including hyperparameter search and testing correctness), leading to total emissions of 5.4kg of $CO_2$. No crowdsourced labor was used in this work. LLM assistants were used for minor stylistic and grammatical corrections of the final manuscript, consistently with ACL recommendations. GitHub Copilot has been used to assist in coding with auto-completion but no script or algorithm has been fully generated with it.

## B  DATASETS AND DATA GENERATION DETAILS

### B.1  DATASETS LIST

We created 6 datasets with a balanced number of fake and true samples (1 dataset per generator). See table 3 for the list of generators used for the datasets and the precise size of the datasets. The datasets are split with 80% for training, 10% for eval (choosing the best model to save), and 10% for testing. Before cutting the samples to the 500 characters threshold, we filtered out samples smaller than 500 characters to only have samples of 500 characters. A small improvement here would be to use the "min_new_token" generation parameter so that we would not need to filter out smaller samples. We only used this parameter to generate the adversarial datasets. The samples of the true articles are the same across all datasets except for the discarded samples.

There is also a 4th auto-complete model dataset, "round-robin," which is a mixture of 2500 samples from each of the other complete models' (Phi-2, Gemma, and Mistral) datasets.

For the adversarial datasets, we reuse these same datasets, but we generate the fake samples with a different prompt or generation parameter (see appendix B.4 for the adversarial prompts). In total, there is one adversarial version of each dataset per attack (i.e. number of datasets times number of attacks adversarial datasets).

| Generator type | Generator | Full Dataset size |
|---|---|---|
| Complete models | Phi-2 | 19918 samples (9960 True/9958 False) |
| | Gemma-2B | 19783 samples (9891 True/9892 False) |
| | Mistral | 19719 samples (9860 True/9859 False |
| | Phi-2/Gemma-2B/Mistral | 8325 samples (4164 True/4161 Fake) |
| Chat models | LLama-3-8B-Instruct | 19917 samples (9960 True/9957 Fake) |
| | Gemma-chat | 19917 samples (9960 True/9957 False) |
| | Zephyr | 17691 samples (8846 True/8845 False) |

Table 3: List of datasets of fake and true articles. Each generator corresponds to 1 dataset.

### B.2  PROMPTS FOR GENERATING DATA

For the auto-complete models, we only use the prefix as the prompt. For chat models, we use the prompt in table 4. Also, we force the first tokens of the output to be the prefix for that particular sample. This prevents the chat model from generating typical assistant-starting messages that would be too obvious to spot for the detectors (in practice, an attacker could also remove it easily). For Gemma-2B-it (Gemma chat), there is no system prompt in the chat template; we simply dropped the system prompt for that case.

| System Prompt | User Prompt |
|---|---|
| You are a helpful assistant. | Continue to write this news article: {prefix} |

Table 4: Prompt used to generate the fake news articles with the chat models.

### B.3  DATASET EXAMPLE

Below is an example of a true and an LLM-generated news article from the Zephyr dataset. As explained in the methodology, samples are regrouped by pairs of true and fake samples (ordered randomly within the pair). The true and fake samples in the pair start with the same 10-word prefix.

**True sample:**
*"Former Vice President Dick Cheney on Sunday defended the Bush administration's economic record, the invasion of Iraq and the treatment of suspected terrorists, warning that reversing its anti-terrorism policies endangers Americans. "We've accomplished nearly everything we set out to do," ex-Vice President Dick Cheney says Sunday about Iraq. In a wide-ranging interview with CNN's "State of the Union," Cheney said the harsh interrogations of suspects and the use of warrantless electronic surveilla"*

source: `https://edition.cnn.com/2009/POLITICS/03/15/cheney.interview/`

**Fake sample:**
*"Former Vice President Dick Cheney on Sunday defended the Bush administration's use of enhanced interrogation techniques, commonly referred to as torture, in the aftermath of the September 11 attacks. Speaking at the Reagan Library's National Security Forum in Simi Valley, California, Cheney argued that the use of waterboarding and other techniques were crucial in obtaining valuable intelligence and preventing further attacks on American soil. "We did learn a lot," Cheney said. "We learned, for e"*

source: `https://huggingface.co/HuggingFaceH4/zephyr-7b-beta`

## B.4    ATTACK PROMPTS

The idea behind the attacks we crafted to generate news articles evading the detectors is to generate news articles with a distribution of words looking more like the CNN news articles that were used to train the detectors. This is particularly true for the "news prompt" attack that asks the model to generate a CNN news-looking article. The same applies to the "example prompt," which uses in-context learning to generate a more CNN news-looking article. The news prompt can be found below in 4 and the example prompt in 7.

For the "tweet prompt," the idea is to generate text with a different distribution than news articles, which might confuse detectors trained on news articles. The "paraphrasing prompt" has a similar effect of modifying the distribution of words to make it more diverse than the original output. The tweet prompt can be found in 5 and the other paraphrasing prompt in 6.

---

**Attack prompt - News prompt**

**System Prompt:** You are an intern news writer for CNN. You write news articles with grammar and spelling errors, like all humans do. Some of your sentences don't make a lot of sense due to unexpected words used.

**User Prompt:** Your job is to write a news article starting with: {prefix}

---

Figure 4: News prompt used to generated CNN news looking news articles.)

---

**Attack prompt - Tweet prompt**

**System Prompt:** You are a tweeter user tweeting news information from news articles to your followers

**User Prompt:** Write a 500 characters news tweet starting with: {prefix}

---

Figure 5: Tweet prompt used to generate news information in a tweet format.)

---

**Attack prompt - Paraphrasing prompt**

**System Prompt:** You are a paraphraser. You are given an input passage 'INPUT'. You should paraphrase 'INPUT' to print 'OUTPUT'." "'OUTPUT' shoud be diverse and different as much as possible from 'INPUT' and should not copy any part verbatim from 'INPUT'." "'OUTPUT' should preserve the meaning and content of 'INPUT' while maintaining text quality and grammar." "'OUTPUT' should not be much longer than 'INPUT'. You should print 'OUTPUT' and nothing else so that its easy for me to parse.

**User Prompt:** INPUT: {fake_text}"

---

Figure 6: Prompt used to paraphrase an LLM-generated news article. The prompt is taken from Sadasivan et al. (2024).

---

**Attack prompt - Example prompt**

**System Prompt:** You will receive an example of news article delimited by <ARTICLE_START> and <ARTICLE_END>. You will write news articles following the same writing style to immitate the original writer's writing.

**User Prompt:** Here is an example of a news article:
<ARTICLE_START>
Dozens more bodies have been recovered from a mass grave at a hospital in Khan Younis, according to the Gaza General Directorate of Civil Defense. The Civil Defense said 324 bodies had now been recovered at the Nasser Medical Complex following the withdrawal of Israeli forces from the area earlier this month. In the latest recovery efforts, the bodies of 51 people of "various categories and ages" had been recovered. Of them, 30 bodies were identified. Col. Yamen Abu Suleiman, Director of Civil Defense in Khan Younis, previously told CNN that some of the bodies had been found with hands and feet tied, and there were signs of field executions. The Civil Defense said Wednesday that crews would continue search and recovery operations in the coming days.
<ARTICLE_END>
Write a news article in the same style as the article above starting with:

---

Figure 7: Example prompt used to generate CNN news-looking news article. The article is taken from `https://edition.cnn.com/articles`.

## C  TRAINING DETAILS

### C.1  HYPERPARAMETERS FOR TRAINING

We present in table 5 the hyperparameters used for training. The training was done either on the Nvidia A100 40GB or on the Nvidia V100 16GB, depending on the batch size, model size, and training method. All the trainings were done with 1 epoch. While we used only the models trained with full finetuning on the main part of the paper, we provide some results with different training methods in appendix E. The hyperparameters were chosen according to the hyperparameters used in the original papers of the different models with some adaptation according to the size of the models and testing that the training converges. A linear schedule with a 10% warmup was applied to the learning rate.

## D  FULL HEATMAP FOR TRAINED DETECTORS

You can find in figure 8 the TPR obtained for all trained detectors when testing them on all the datasets (it is a more complete version of figure 2). This plot uses the same metric as the main paper plots, i.e. we find thresholds on the eval set such that we target an FPR of at most 5 %.

| Model | Training method | Hyperparameters | |
|---|---|---|---|
| | | Learning rate | Batch size |
| Distil-RoBERTa | Freeze base | 1e-3 | 64 |
| | Adapter | 3e-4 | 16 |
| | Full Finetuning | 3e-5 | 16 |
| RoBERTa-Large | Freeze base | 1e-3 | 64 |
| | Adapter | 1e-4 | 16 |
| | Full Finetuning | 1e-5 | 16 |
| Electra-Large | Freeze base | 1e-3 | 64 |
| | Adapter | 1e-4 | 8 |
| | Full Finetuning | 1e-5 | 16 |

Table 5: Hyperparameters used for training the detectors. We usually set the batch size to the largest possible for full-finetuning on a Nvidia A100 40GB.

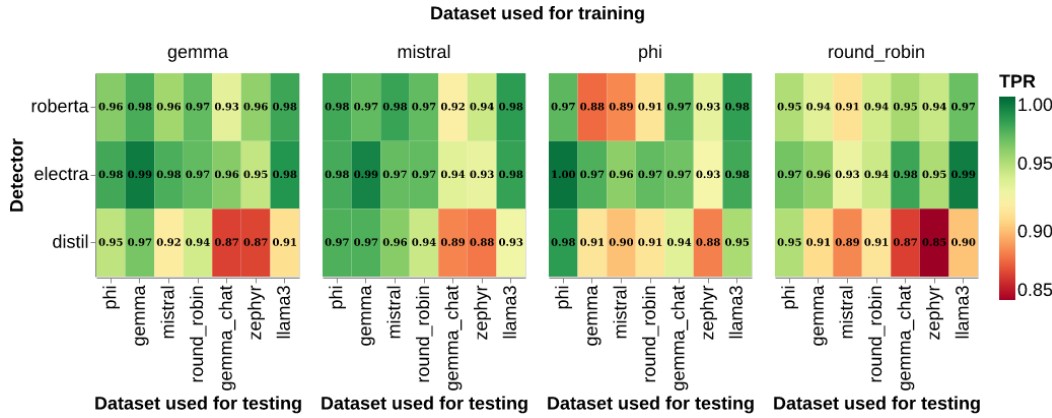

Figure 8: TPR comparison of detectors tested on dataset generated by the chat models.

# E    ROC AUC SCORE WITH DIFFERENT TRAINING METHODS

While we mainly tested trained detectors with full finetuning, we also tested and compared the results using different training methods. First, we tested freezing the LLM detector model's parameters and finetuning the classifier head (freeze base). Secondly, we tested finetuning the base LLM detector with the adapters PEFT method (see Houlsby et al. (2019)). The results we show in this section are the same experiment as presented above in appendix D and in figure 9. However, we decide to use ROC-AUC as the metric to have a threshold-independent metric, avoiding finding a suitable threshold for each case. Also, we compare this ROC-AUC score obtained with freeze base (figure 9) and adapters (figure 10) method to the full-finetuning ROC-AUC (figure 11).

We find that training with the adapters method yields results similar to training with full-finetuning, as expected. This gives an interesting alternative to train detectors.

## E.1    TRAINING WITH FINETUNING ONLY THE CLASSIFICATION HEAD

See figure 9.

## E.2    TRAINING ADAPTER METHOD

See figure 10.

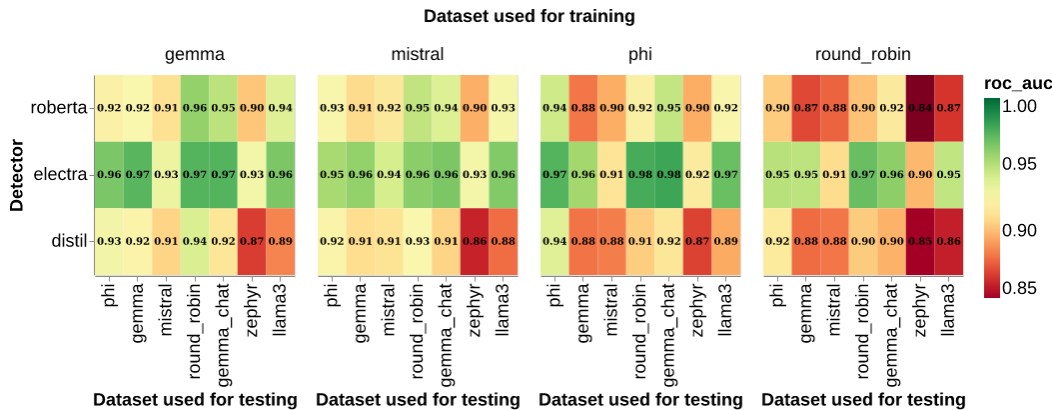

Figure 9: ROC-AUC comparison of detectors tested on dataset generated by the chat models. Detectors are trained by only finetuning the classification head

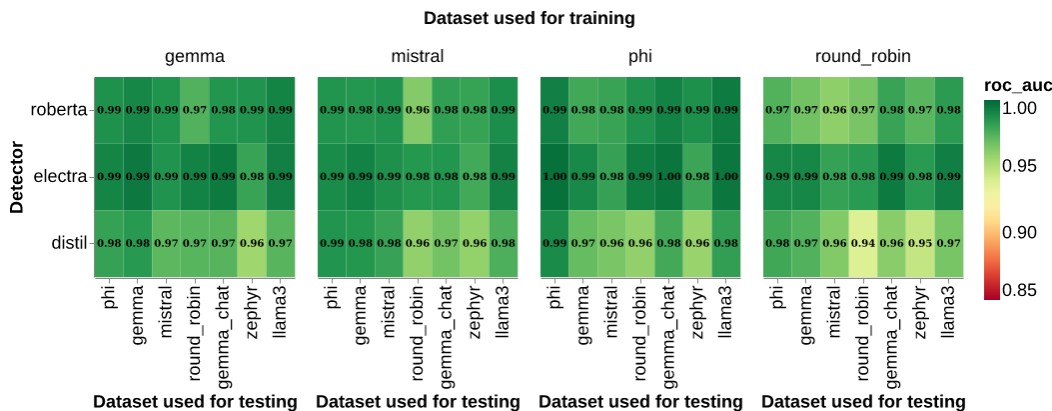

Figure 10: ROC-AUC comparison of detectors tested on dataset generated by the chat models. Detectors are trained with the adapter PEFT method

### E.3 TRAINING WITH FULL FINETUNING

See figure 11.

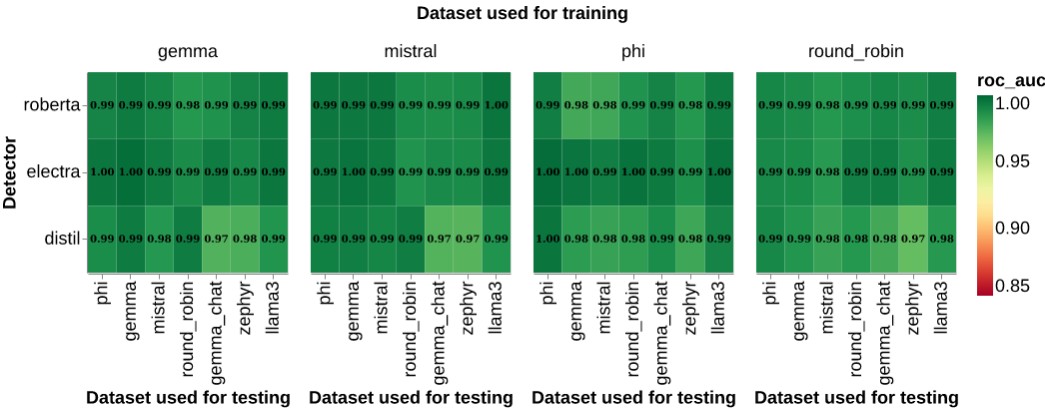

Figure 11: ROC-AUC comparison of detectors tested on dataset generated by the chat models. Detectors are trained with full finetuning

## F    CHECKING DETECTOR DEGRADATION ON MLM TASK

To compare the effect of different finetuning methods on the trained detectors, we evaluate the loss on the MLM (Masked Language Model) task before and during training. In particular, we use samples from `https://huggingface.co/datasets/Polyglot-or-Not/Fact-Completion?row=0`, a dataset of knowledge facts where the task is to fill the gap. This enables us to test if the base models "forgotten facts" during the finetuning. This is shown in figure 13 with the degradation loss that shows how the loss on this MLM task evolves during training. We also provide a baseline where the gap is filled by a model without pre-trained weights. As shown in the figure, the degradation increases at the beginning of training and then stays almost constant. By comparing with figure 12, the eval loss stops improving at the same moment the degradation loss stops increasing, showing signs of convergence. Also, we noticed that the degradation was slightly lower when finetuning with the adapters using the PEFT method. However, the degradation difference between the two training methods might be higher with different tasks and training hyperparameters.

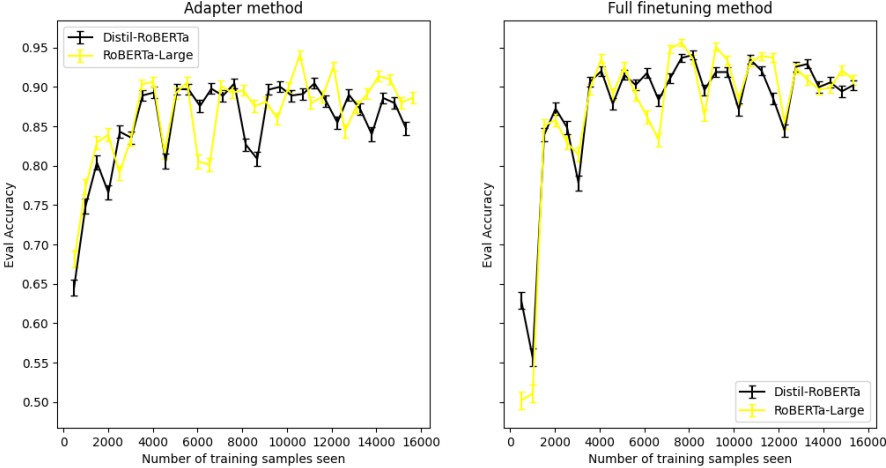

Figure 12: Evaluation loss during training on Mistral generated news samples for Distil-RoBERTa and RoBERTa-Large with Adapters PEFT method and Full finetuning.

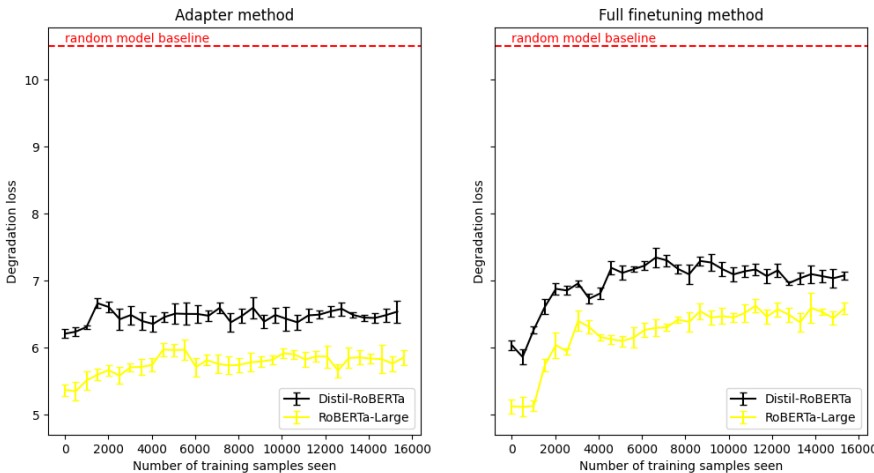

Figure 13: Loss on the MLM task during training for Distil-RoBERTa and RoBERTa-Large with Adapters PEFT method and Full finetuning.

# G   FULL ATTACK TESTING RESULTS

We provide here the same results as in 1, but with data also generated by Gemma-2b-it (Gemma-Chat) and Llama-3 Instruct (7B). As shown in table 6, attacks using Gemma-Chat are less effective. We find that the attacks are often on par with the effectiveness using Zephyr, sometimes above, sometimes below (in terms of TPR difference).

| Generator | Detector | No Attack | High temperature | Repetition penalty | Attack News prompt | Tweet prompt | Paraphrasing | Example prompt |
|---|---|---|---|---|---|---|---|---|
| Gemma-Chat | Electra_Mistral | 93% ± 1% | (↓11%) 82% ± 2% | (↓7%) 86% ± 2% | (↓3%) 90% ± 1% | (↓7%) 86% ± 2% | (↑4%) 97% ± 1% | (↑1%) 94% ± 1% |
| | Electra_RR | 98% ± 0.4% | (↓4%) 94% ± 1% | (↓2%) 96% ± 2% | (↓3%) 96% ± 1% | (↑1%) 99% ± 0.5% | (↑2%) 99.6% ± 0.3% | (↓0%) 98% ± 1% |
| | RoBERTa OpenAI | 80% ± 1% | (↓12%) 68% ± 2% | (↓29%) 51% ± 2% | (↓4%) 76% ± 1% | (↓32%) 48% ± 2% | (↓4%) 76% ± 2% | (↓19%) 61% ± 02% |
| | Fast-DetectGPT | 94% ± 1% | (↓23%) 71% ± 2% | (↓40%) 54% ± 2% | (↓3%) 93% ± 1% | (↓20%) 74% ± 2% | (↓8%) 86% ± 2% | (↓20%) 74% ± 2% |
| Llama-3 Instruct | Electra_Mistral | 98% ± 0.4% | (↓10%) 88% ± 2% | (↓31%) 67% ± 2% | (↓14%) 84% ± 2% | (↓16%) 82% ± 2% | (↓23%) 75% ± 2% | (↓1%) 97% ± 1% |
| | Electra_RR | 99% ± 0.2% | (↓4%) 95% ± 1% | (↓14%) 85% ± 2% | (↓11%) 88% ± 1% | (↓7%) 92% ± 1% | (↓6%) 93% ± 1% | (↓1%) 98% ± 1% |
| | RoBERTa OpenAI | 76% ± 1% | (↓16%) 60% ± 2% | (↓43%) 33% ± 2% | (↓48%) 28% ± 2% | (↓43%) 33% ± 2% | (↓8%) 68% ± 2% | (↓5%) 71% ± 2% |
| | Fast-DetectGPT | 97% ± 1% | (↓14%) 83% ± 2% | (↓63%) 34% ± 2% | (↓14%) 83% ± 2% | (↓22%) 75% ± 2% | (↓36%) 61% ± 2% | (↓4%) 93% ± 1% |
| Zephyr | Electra_Mistral | 93% ± 0.8% | (↓16%) 77% ± 2% | (↓34%) 59% ± 2% | (↓3%) 90% ± 1% | (↓35%) 58% ± 2% | (↓22%) 71% ± 2% | (↓3%) 90% ± 1% |
| | Electra_RR | 95% ± 0.8% | (↓7%) 88% ± 1% | (↓10%) 85% ± 2% | (↑1%) 96% ± 1% | (↓11%) 84% ± 2% | (↓1%) 94% ± 1% | (↓2%) 93% ± 1% |
| | RoBERTa OpenAI | 57% ± 2% | (↓14%) 43% ± 2% | (↓31%) 26% ± 2% | (↑5%) 62% ± 2% | (↓33%) 24% ± 2% | (↓0%) 57% ± 2% | (↓3%) 54% ± 2% |
| | Fast-DetectGPT | 86% ± 1% | (↓32%) 54% ± 2% | (↓62%) 24% ± 2% | (↓2%) 84% ± 2% | (↓44%) 42% ± 2% | (↓41%) 45% ± 2% | (↓3%) 83% ± 2% |

Table 6: TPR obtained when testing the detectors on the adversarial generated test set with the threshold to target an FPR of at most 5%. Electra_Mistral is Electra-Large finetuned on Mistral samples, and Electra_RR is Electra-Large finetuned on the Round-Robin dataset we created (the main paper table shows the results for Zephyr generated advesarial text only).

# H   MODELS, DATASETS AND THIRD-PARTY CODE

| Name | Retrieved From |
|---|---|
| RoBERTa-Large | huggingface link: `https://huggingface.co/FacebookAI/roberta-large` |
| Distil-RoBERTa | huggingface link: `https://huggingface.co/distilbert/distilroberta-base` |
| Electra-Large | huggingface link: `https://huggingface.co/google/electra-large-discriminator` |
| RoBERTa detector from OpenAI | huggingface link: `https://huggingface.co/openai-community/roberta-base-openai-detector` |
| Phi-2 | huggingface link: `https://huggingface.co/microsoft/phi-2` |
| Gemma-2B | huggingface link: `https://huggingface.co/google/gemma-2b` |
| Mistral | huggingface link: `https://huggingface.co/mistralai/Mistral-7B-v0.1` |
| Gemma-chat | huggingface link: `https://huggingface.co/google/gemma-2b-it` |
| Zephyr | huggingface link: `https://huggingface.co/HuggingFaceH4/zephyr-7b-beta` |
| LLama-3-8B-Instruct | huggingface link: `https://huggingface.co/meta-llama/Meta-Llama-3-8B-Instruct` |
| CNN Dailymail | huggingface link: `https://huggingface.co/datasets/cnn_dailymail` |
| Fast-DetectGPT | GitHub link: `https://github.com/baoguangsheng/fast-detect-gpt/blob/main/scripts/fast_detect_gpt.py` |
| Xsum dataset (BBC News) | huggingface link: `https://huggingface.co/datasets/EdinburghNLP/xsum` |

Table 7: Urls from which models and code were retrieved

