# OpenReview forum: "LLM Detectors Still Fall Short of Real World: Case of LLM-Generated Short News-Like Posts"
_ICLR.cc/2025/Conference — Submitted to ICLR 2025_

### Official Review · Reviewer_LMFw · 2024-10-27

**Soundness:** 3
**Presentation:** 3
**Contribution:** 2
**Rating:** 5
**Confidence:** 4

**Summary:**

This paper investigates the effectiveness of various LLM text detectors on short news-like posts on social media. It is shown that existing zero-shot detectors are easily fooled by trivial attacks, such as increasing the temperature or repetition penalty during the LLM text generation process. Detectors which are trained on specific LLMs may generalize to new LLMs but fail on off-distribution human-written texts. This suggests that current LLM detector benchmarks are insufficient and highlights the need for dynamic benchmarks that can be adapted to specific domains and threat models.

**Strengths:**

1. Evaluations on off-distribution human texts which was missing in prior literature
2. It is interesting to see that some zero-shot detectors do not generalize to temperature increases/repetition penalty

**Weaknesses:**

1. Limited novelty: There are no new insights or technical contributions offered in the paper, and many of the results are not surprising. It is well-known that zero-shot detectors can be very sensitive to distribution shifts (which is cited by the authors) and purpose-trained detectors may be overfit to their training distribution.

2. The authors do not measure or control for quality degradation and/or content preservation after the attacks. It is hard to make sense of the results as the attacker has a lot of power without these constraints. For example, higher temperature may make the model less fluent/coherent which may make it an impractical attack

**Questions:**

I think it would be very interesting to explore the trade-off between generalization to off-distribution LLMs vs off-distribution human texts in more detail. An interesting direction here would be to identify subpopulations in human text which are prone to be classified as LLM generated, or vice versa. This would shed more light on when these detectors may fail and how they may be biased towards or against certain human groups or LLMs.

---

> ### Author Response · Authors · 2024-11-28
> **Author's response**
>
> We thank the reviewer for their feedback.
>
> We believe that some of the weaknesses highlighted by the reviewer stem from our lack of clarity regarding our contribution and scope, which we have clarified in the revised manuscript.
>
> As we mention in our response to other reviewers, the heart of our contribution is showing that the methodology of the prior LLM benchmarks is flawed in principle, leading to a trivial evasion attack still being effective. The LLM detectors evaluated as SotA in third-party benchmarks (notably RAID [1]), significantly underperforming and not being ready for real-world use.
>
> Specifically,
>
> 1. We show that large-scale static benchmarks, such as RAID [1], are not suited for evaluating LLM detectors in a way relevant to the real-world setting. This issue is due to:
>  1.1 LLM-generated text detection is inherently happening in adversarial settings, with the attacker seeking to evade detection. As such, reference static benchmarks provide a baseline not only for the LLM detector developers but also for attackers, who will learn to evade them. This adversarial setting radically differs from the general LLM evaluation practices, where benchmarks are useful to validate the model's capabilities. While we are the first ones to state this in the context of LLM detection, this difference in evaluations of ML solutions in adversarial settings has been previously highlighted, notably in cybersecurity [2]. In particular, the failure of recent benchmarks to test for temperature increase attacks led to all detectors being vulnerable to it, constituting an example of such an attack
>  1.2 LLM detectors' performance varies wildly depending on the task, even for closely related tasks, making large-scale benchmarks non-useful even for the best detector selection task, even of highly related topics, such as ours and "news articles" in the RAID benchmark [1], including for identical attacks, as highlighted in 4.3 "Prior Benchmarks" section.
>
> 2. We show that all the previously published benchmarks on LLM detectors are flawed in principle because they do not consider the tradeoff between the generalization to previously unseen attacks (TPR generalization) and previously unseen types of human-written texts (FPR generalization). While this tradeoff is intuitive, it has neither been stated in the LLM detectors literature nor explicitly benchmarked as a tradeoff in the past, leading to real-world issues with LLM detectors FPR generalization, notably false positives on text written by non-native writers [3].
>
> 3. We propose a method to test LLM detectors that is useful for evaluating their real-world performance, that can be dynamically expanded, and provide code to achieve it. Specifically, we do not provide a dataset to evaluate LLM detectors on, but a method to derive an evaluation dataset from samples of texts in an application scope, given a threat model, which is radically different from prior work in the field, that instead attempted to build universal, scope and threat model-agnostic datasets.
>
> We hope that this addresses the weakness W1 stated by the reviewer.
>
> We agree with the reviewer's point in W2. However, prior work incorporating perplexity analysis and human evaluation has shown a minimal text quality degradation, even for the strongest of the attacks considered - paraphrasing [4]. As such, we did not perform the perplexity analysis for attacks we considered, but it is definitely a point to consider. We thank the reviewer and have added a point about it to the methodology and discussion section.
>
> Similarly, we agree with the reviewer that a study of human-written texts is often misclassified as LLM-generated would be highly interesting. We have anecdotal reports of neurodivergent individuals being most susceptible to such misclassification (personal communication with Prof. Dr. Rua Mae Wiliams, PhD), but to our knowledge, there is no such systemic study, and we are not aware of datasets that could be used for it.
>
> Again, we thank the reviewer for their feedback and apologize for the lack of clarity in our original manuscript. We adjusted the manuscript to respond to the reviewers' remarks and performed an additional pass to correct grammatical errors and citation usage inconsistencies in the main text.
>
> [1] Dugan, Liam et al. "RAID: A Shared Benchmark for Robust Evaluation of Machine-Generated Text Detectors." Annual Meeting of the Association for Computational Linguistics (2024)
>
> [2] Arp, Daniel et al. "Dos and don'ts of machine learning in computer security." USENIX Security 31 (2022)
>
> [3] Liang, Weixin et al. "GPT detectors are biased against non-native English writers." Patterns 4 (2023)
>
> [4] Sadasivan, Vinu Sankar et al. "Can AI-Generated Text be Reliably Detected?" ArXiv abs/2303.11156 (2023)

---

> ### Comment · Reviewer_LMFw · 2024-11-30
> **Agree broadly with your points, but execution is lacking**
>
> Thank you for the rebuttal. In general, I'm in broad agreement with the ideas advanced in the work, and it is consistent with my own findings in this space. However, as acknowledged by you, the ideas and contributions are not particularly novel, and have been brought up before. In such a scenario, to merit acceptance, the paper must be well executed. As other reviewers have mentioned, it would be good to conduct these experiments on a large scale, either from the angle of poor generalization across human texts (you would need to collect some data for this) or just increasing the scope of detectors/LLMs considered in the paper.

---

> > ### Author Response · Authors · 2024-12-02
> > **Thank you for actionable recommendations**
> >
> > Thank you for your response; we appreciate the broad agreement with our ideas and actionable recommendations for improving the paper.
> >
> > Once again, while the ideas are somewhat intuitive and have been presented in other fields, we have not seen them formulated in the context of LLM detectors development and benchmarking, which, in turn, led to poor defensive tools benchmarking practices and a false sense of security in that field. Seeing that, we opted to publish a single investigation of real-world relevance to avoid yet another poorly defined benchmark without insights relevant to its real-world usability.

---

### Official Review · Reviewer_Q7Dz · 2024-10-29

**Soundness:** 2
**Presentation:** 1
**Contribution:** 2
**Rating:** 3
**Confidence:** 3

**Summary:**

This paper studies LLM generated content detection via (LLM-based) detectors. Zero-shot detectors are found to be vulnerable to simple evasion attacks, while a manually trained detector can overcome this issue but failed to not flag human-written texts as LLM generated.

**Strengths:**

- The studied question is highly practical and is crucial for societal concerns. Exploring the limitation of LLM generated content detector can help flag early signals on the boundary of such event and help guide better design choices for future content detectors.
- The found effectiveness using simple evasion attack on zero-shot detectors is important, suggesting the need of improvement when constructing benchmarks for generated content detection.

**Weaknesses:**

The weaknesses I spot are as follows.
- The coverage of experiments is a bit narrow. All experiments studied in this paper are totally based on the CNN Dailymail news dataset and its variants generated by LLMs. This dataset bias could lead to biased results as well. The study should incorporate a broader spectrum of candidate dataset types.
- The claim that trained detectors fail to generalize to unseen human-written texts needs further evidence. As illustrated in the paper, the Electra_RR model benefits from a dataset mixture training for improving TPR. This trend is not discussed for FPR, thus the failure mode on US vs British English detection may be a fact of lacking such data in the training procedure and the 5% FPR location step, and not because trained detectors are not capable of doing so.
- The writing could be improved. There are grammar issues and incomplete sentences throughout the paper. Besides, the misuse of \citet and \citep makes the paper hard to read.

**Questions:**

- Will the dataset mixture training statement hold true for FPR? Can the failure in Table 2 be solved by incorporating both US and British style human-written texts during training, or by selecting a proper threshold using both styles of texts?
- Will the claims hold on datasets other than news? For example, on books or review similar to the categorization in [1]?

[1] RAID: A Shared Benchmark for Robust Evaluation of Machine-Generated Text Detectors

---

> ### Author Response · Authors · 2024-11-28
> **Author's response**
>
> We thank the reviewer for their feedback.
>
> We believe that some of the weaknesses highlighted by the reviewer stem from our lack of clarity regarding our contribution and scope, which we have clarified in the revised manuscript.
>
> We understand the W1 remark by the reviewer that the experiment coverage could be improved by including additional human reference datasets in addition to CNN Dailymail. However, we believe that the failure of standard methodologies for LLM detector testing we demonstrate and the demonstration of the failure of detectors performing best according to third-party benchmarks is sufficient to make our point both as to the readiness of LLM detectors in the real-world use case in information operations detection, as well as general methodological issues with current state with LLM detectors testing.
>
> We agree with the reviewer's statement in W2 that the ELECTRA-RoundRobin would perform better if fine-tuned on more diverse human samples. However, the point of this experiment and Table 2 is to demonstrate a methodological flaw in prior approaches. Specifically, we show that there is a tradeoff between the generalization to previously unseen evasion attacks (TPR generalization) and previously unseen types of human-written texts (FPR generalization), which has so far never been evaluated as such in LLM detector benchmarking papers, despite leading to real-world harm [1]. Once again, the goal of ELECTRA-RoundRobin is to show that fine-tuning a classifier for detection - which is still commonly reported in LLM detector benchmarks as achieving SotA - can lead to great performance in adversarial setting at the cost of reference human dataset overfitting, which in turn is not evaluated. However, if we were attempting to develop a novel SotA detector, including diverse samples of human writing would have indeed been essential.
>
> We thank the reviewer for highlighting the issues with the quality of writing (W3) and have performed an additional pass for grammar correctness and consistent usage of /citet /citep.
>
> We believe our response to W2 also responds to question Q1. While it is possible to build better LLM detectors by increasing the diversity of human text samples, our goal is to show that the performance of LLM detectors needs to be evaluated on the human text samples in the correct subject domain.
>
> We thank the reviewer for the question Q2. We compared our results with the RAID benchmark [2] by matching their "news" category, generators, detectors, and attacks that were identical but obtained significantly different detection rates, as highlighted in the 4.3 "Prior Benchmarks" section. While we focused on generated news-like content as most socially damaging in the immediate, our method also applies to other domains, assuming a threat model is defined. While the generated reviews are likely to be sufficiently similar to share the threat model and length limitations, books are significantly longer. They will likely have more extensive prompts and undergo segmental rewrites or regenerations. In that latter case, generation and evasion procedures will need to be re-implemented in a way consistent with the considered threat model before conclusions can be drawn. While the conclusion regarding their detectability might differ from the short news-like content generation, the methodological contribution of our work will still apply - namely, the need for a scope, threat model, and the evaluation of the tradeoff in generalization on unseen evasion attacks out-of-distribution human-written text samples.
>
> Again, we thank the reviewer for their feedback and apologize for the lack of clarity in our original manuscript. We adjusted the manuscript to respond to the reviewers' remarks and performed an additional pass to correct grammatical errors and citation usage inconsistencies in the main text.
>
> [1] Liang, Weixin et al. "GPT detectors are biased against non-native English writers." Patterns 4 (2023)
> [2] Dugan, Liam et al. "RAID: A Shared Benchmark for Robust Evaluation of Machine-Generated Text Detectors." Annual Meeting of the Association for Computational Linguistics (2024).

---

### Official Review · Reviewer_9DAR · 2024-11-02

**Soundness:** 2
**Presentation:** 2
**Contribution:** 2
**Rating:** 3
**Confidence:** 3

**Summary:**

The paper presents an evaluation on the performance of existing LLM detectors in fake news detection. Both purpose-trained detectors and zero-shot detectors are benchmarked with the newly proposed datasets. They are tested on normal datasets, cross-domain generalization and adversarial evasion techniques.

**Strengths:**

1. Fake news detection is a crucial issue in society.
2. A comprehensive review on the literature is provided.
3. New datasets for fake news detection are generated to evaluate existing LLM detectors.

**Weaknesses:**

1. The presentation is ambiguous and flawed. The first section of introduction does not clearly explain the motivation, problem and method of this paper. Besides, there are some grammatical mistakes in the main text, making it harder to comprehend.
2. This work offers some observations in a real-world scenario of fake news detection, but it lacks insights or findings that can further promote the development of LLM detectors.
3. The experiments only take several small LLMs to generate fake news, which is an assumption of "moderately sophisticated attacker". However, the attacker can also adopt proprietary APIs, e.g., ChatGPT, to generate better information. The detection performance on text generated by more powerful models should be included.

**Questions:**

My major concern is about the position of this paper. It is more like an evaluation of the application in a real world scenario of fake news detection. There is no fundamental problem or challenge in the field presented and addressed in the work. I am not sure if it's suitable to accept it to a conference like ICLR.

---

> ### Author Response · Authors · 2024-11-28
> **Author's response**
>
> We thank the reviewer for their feedback.
>
> We have adjusted the introduction to better respond to the W1 identified by the reviewer and comments from the other reviewers. Similarly, we have corrected the grammatical mistakes we could find.
>
> We respectfully disagree with the evaluation of the reviewer in the W2 that our work is limited to a single observation and lacks impact on the field, given that we believe that the methodological contribution we provide in this paper is major. Specifically, we show that the LLM detector evaluation that has been performed until now is flawed in its real-world relevance, and we propose a new testing methodology for which we validate and provide code.
>
> Specifically, we show that:
>  - The LLM-generated text detection is inherently happening in an adversarial setting, with the attacker seeking to evade detection. As such, reference static benchmarks provide a baseline not only for the LLM detectors but also for attackers, who will learn to evade them. This setting radically differs from the general LLM evaluation, where benchmarks are useful to validate the model's capabilities. While we are the first ones to state this in the context of LLM detection, this difference in evaluations of ML solutions in adversarial settings has been previously highlighted, notably in cybersecurity [2]. We show that this vulnerability is not merely theoretical by demonstrating the effectiveness of a simple temperature-based evasion attack that has been omitted from recent LLM detector benchmarks.
>  - The LLM detector performance varies wildly depending on the task, even for closely related tasks, making large-scale benchmarks non-useful even for the best detector selection task, even of highly related topics, such as ours and "news articles" in the RAID benchmark [1]; highlighted in 4.3 "Prior Benchmarks" section. Similarly, the tolerance to FNR and FPR varies wildly between the fields and should not be compared. Falsely accusing a student of academic misconduct has a radically different impact than not algorithmically promoting a social media post that looks too much like generated text.
>  - All the previously published benchmarks on LLM detectors are flawed in principle because they do not consider the tradeoff between the generalization to previously unseen evasion attacks (TPR generalization) and previously unseen types of human-written texts (FPR generalization). While this tradeoff is intuitive, it has neither been stated in the LLM detectors literature nor explicitly benchmarked as a tradeoff in the past, leading to real-world LLM detectors issues with FPR generalization, notably false positives on text written by non-native writers [2].
>
> We hope that this conceptual contribution to the field in the testing methodology alleviates the concerns about the paper's suitability for the ICLR conference stated by the reviewer in question Q1.
>
> Finally, we also understand the reviewers' suggestion in W3 to extend the evaluation of the detectors to SotA commercial API-only models. We apologize for failing to mention in our threat model that we assume such API providers will be monitoring for and blocking disinformation operations using their models, such as OpenAI recently reported doing [3].
>
> Again, we thank the reviewer for their feedback and apologize for the lack of clarity in our original manuscript. We hope that our response fully alleviated the reviewer's concerns. To prevent any such confusion in the future, we have revised the 1.4 "Contributions" section to reflect better what we stated in the 5 "Conclusion" section and adjusted our threat model presentation to respond to W3. We have performed an additional pass to correct grammatical errors we could find in the main text.
>
> [1] Dugan, Liam et al. "RAID: A Shared Benchmark for Robust Evaluation of Machine-Generated Text Detectors." Annual Meeting of the Association for Computational Linguistics (2024)
>
> [2] Arp, Daniel et al. "Dos and don'ts of machine learning in computer security." USENIX Security 31 (2022)
>
> [3] Metz, Cade "OpenAI Says It Disrupted an Iranian Misinformation Campaign" The New York Times 16.08.2024 (2024)

---

### Official Review · Reviewer_52s7 · 2024-11-02

**Soundness:** 1
**Presentation:** 2
**Contribution:** 1
**Rating:** 3
**Confidence:** 4

**Summary:**

This work evaluates some of the existing LLM-generated short-text detectors, including the zero-short and purpose-trained detectors, on a newly proposed short news-like generated post dataset (with attacks). This work conducted some experiments that revealed that the THREE zero-short detectors investigated show heavily degraded performance when encountering attacks, particularly for a simple attack by increasing LLM sampling temperature. Given these findings, the authors conclude current LLM detectors still fall short of real-world usage.

In summary, this work contributes to a new short news-like dataset, involving three types of evasion attacks, and evaluating some of the existing detectors on this dataset. Finally, this work draws some conclusions based on their empirical findings.

**Strengths:**

- A news-like dataset generated by six LLMs was developed in this work. The source models include three non-chat models, i.e., Phi-2, Gemma-2B, Mistral-0.1; and three chat models, i.e., Gemma-chat, Zephyr, and LLama-3-8B-Instruct. Regarding the generated text, the authors proposed to cut and take only the first 500 characters to reduce detection effects in terms of text length.

- Three different types of attacks, i.e., changing generation parameters, conducting prompting attacks, and paraphrasing attacks were considered to evaluate some of the existing methods. Experiments show detectors evaluated show much degraded detection performance.

**Weaknesses:**

The major weakness is that this work lacks substantial contribution, both in terms of technical contribution and surprising experimental observations or findings. Please see my detailed comments below.

- While a news-like dataset was proposed, the scope of this dataset is rather limited. Since a real-world (as noted in the paper title) application should not be confined to news only -- they can be more broadly e.g. story writing, academic writing, etc. Thus, the findings (and corresponding claims) may probably not hold anymore in a broader scope. A good benchmark should include as diverse subjects as possible (as we evaluate LLMs). I would suggest the authors refer to detection-related works for commonly used benchmarks, e.g. Fast DetectGPT and [1-5].

- Besides, the text length has been restricted to 500 characters, a rather short piece of text. Though this setting was intended to alleviate the impacts of text length in fake text detection, the real-world application, however, a large of amount of generated text can be rather long texts. As a result, the conclusion drawn from this dataset may be applicable to long-text detection scenarios. Indeed, short-short detection is much harder than their long text counterpart. Unfortunately, some methods specifically designed for short-text detection are not discussed or evaluated. As a good benchmark dataset, it is strongly encouraged to be thorough, and objectively reflecting different detection scenarios.

- The source models generating fake texts are far from enough, as you are considering the real-world usage, more advanced models, such as GPT3.5 Turbo, GPT 4, GPT 4o, Claude etc, are strongly supposed to be considered (because these are better performing models tend to be preferred in real-world applications). Otherwise, the benchmark may not be representative enough. Please refer to more recent literature (e.g. [1-5], and follow generally-adopted routines to include such advanced models.

- This work lacks a thorough literature review, the methods evaluated are far from enough to draw the conclusion that "LLM detectors still fall short of the real world", because many recent state-of-the-art detection methods have been missing! To make this claim and findings more solid, I suggest additionally considering [1-5] as your baseline methods.

- In fact, all machine learning models can be vulnerable to attacks, this may not prevent their usage in the real-world scenario. For example, the existence of adversarial examples (in face recognition, object detection etc), a typical type of evasion attack, does not prevent the wide application of these models. In other words, I don't find the robustness issues (including some attacks mentioned in this work) a surprising finding, given that many of these existing works also reported their drawbacks to these attacks! This is a widely-known problem, rather than first reported in this work! As a result, this can limit the novelty of this work.

[1] Yang, Xianjun, Wei Cheng, Yue Wu, Linda Petzold, William Yang Wang, and Haifeng Chen. "Dna-gpt: Divergent n-gram analysis for training-free detection of gpt-generated text." ICLR 2024.

[2] Mao, Chengzhi, Carl Vondrick, Hao Wang, and Junfeng Yang. "Raidar: geneRative AI Detection viA Rewriting." ICLR 2024.

[3] Tian, Y., Chen, H., Wang, X., Bai, Z., Zhang, Q., Li, R., Xu, C. and Wang, Y., 2023. Multiscale positive-unlabeled detection of ai-generated texts. ICLR 2024.

[4] Hans, Abhimanyu, Avi Schwarzschild, Valeriia Cherepanova, Hamid Kazemi, Aniruddha Saha, Micah Goldblum, Jonas Geiping, and Tom Goldstein. "Spotting llms with binoculars: Zero-shot detection of machine-generated text." ICML 2024.

[5] Su, J., Zhuo, T.Y., Wang, D. and Nakov, P., 2023. Detectllm: Leveraging log rank information for zero-shot detection of machine-generated text. arXiv preprint arXiv:2306.05540.

**Questions:**

For my detailed comments, please see the weakness part. I suggest the authors clearly state the major contributions and findings in the rebuttal, e.g. what are major findings that were not mentioned in previous works, what are major differences of the proposed dataset, compared with benchmark datasets used by many existing fake tex detection works.

**Details Of Ethics Concerns:**

NA.

---

> ### Author Response · Authors · 2024-11-28
> **Author's response - 1/2**
>
> We thank the reviewer for their feedback.
>
> While the provided feedback is relevant, we believe that the weaknesses stated by the reviewer might stem from our failure to convey the core contributions of our work sufficiently clearly.
>
> Specifically, in response to the reviewer's request for a contribution summary, the heart of our contribution is showing that the methodology of the prior LLM benchmarks is flawed in principle.
>
> Specifically,
>
> 1. We show that large-scale static benchmarks, such as RAID [1], are not suited for evaluating LLM detectors in a way relevant to the real-world setting. This issue is due to:
>   - LLM-generated text detection is inherently happening in adversarial settings, with the attacker seeking to evade detection. As such, reference static benchmarks provide a baseline not only for the LLM detector developers but also for attackers, who will learn to evade them. This adversarial setting radically differs from the general LLM evaluation practices, where benchmarks are useful to validate the model's capabilities. While we are the first ones to state this in the context of LLM detection, this difference in evaluations of ML solutions in adversarial settings has been previously highlighted, notably in cybersecurity [2]. In particular, the failure of recent benchmarks to test for temperature increase attacks led to all detectors being vulnerable to it, constituting an example of such an attack
>   -  LLM detectors' performance varies wildly depending on the task, even for closely related tasks, making large-scale benchmarks non-useful even for the best detector selection task, even of highly related topics, such as ours and "news articles" in the RAID benchmark [1], including for identical attacks, as highlighted in 4.3 "Prior Benchmarks" section.
>
> 2. We show that all the previously published benchmarks on LLM detectors are flawed in principle because they do not consider the tradeoff between the generalization to previously unseen attacks (TPR generalization) and previously unseen types of human-written texts (FPR generalization). While this tradeoff is intuitive, it has neither been stated in the LLM detectors literature nor explicitly benchmarked as a tradeoff in the past, leading to real-world issues with LLM detectors FPR generalization, notably false positives on text written by non-native writers [3].
>
> 3. We propose a method to test LLM detectors that is useful for evaluating their real-world performance, that can be dynamically expanded, and provide code to achieve it. Specifically, we do not provide a dataset to evaluate LLM detectors on, but a method to derive an evaluation dataset from samples of texts in an application scope, given a threat model, which is radically different from prior work in the field, that instead attempted to build universal, scope and threat model-agnostic datasets.
>
> Overall, our contribution is methodological, first and foremost, for the machine learning field. We focus on a narrow application scope (generated short news-like content) and a specific threat model (moderately sophisticated attacker), and we make sure that we develop an adapted testing methodology. This deep dive focuses on getting the methodology right and differs from prior benchmarking works that attempted to cover as many models, attacks, and topics as possible; an approach that we believe, for reasons stated above, is flawed.
>
> We hope that this makes it clear to the reviewer why we disagree with their W1 suggestion to use common benchmark datasets in principle, given the methodological flaws of that approach stated above.
>
> Similarly, in the context of not presenting contributions 1-3 above sufficiently clearly in our original manuscript, we understand the reviewer's question regarding the choice of text length stated by the reviewer in W2. However, once again, for methodological reasons above, an evaluation of LLM detectors translating to real-world performance requires a choice of a specific scope. Consistently with prior literature on the potential for negative impact of publicly accessible powerful LLMs, we chose the news-like texts on social media in information operations, opting to optimize the methodology in that context rather than trying to be overly general. Once again, we do not claim to contribute a new LLM detector evaluation dataset, nor do we attempt a universal evaluation.

---

> > ### Author Response · Authors · 2024-11-28
> > **Author's response - 2/2**
> >
> > We understand as well the reviewers' suggestion in W3 to extend the evaluation of the detectors to SotA commercial API-only models, and we apologize for not failing to mention in our threat model that we assume such API providers will be monitoring for and blocking disinformation operations using their models, such as OpenAI recently reported doing [4].
> >
> > While we understand the reviewer's suggestion in W4 to evaluate more detectors, we do not believe it is feasible or necessary. With the constant publication of novel LLM detectors, there will always be detectors we did not cover, given that they did not exist, did not provide working code, or were not considered sufficiently well-known to be tested at the moment of experiments, as opposed to the time of review. Specifically, here, most of the detectors cited by the reviewer were published and presented in mid-2024, after we finished the experimental part of this work. Similarly, due to the paper size limitation, we had to select the detectors we presented in the background and prior work sections. Second, we focus on top-performing detectors in third-party benchmarks, notably RAID [1]. We found no reason to believe that detectors that did not score significantly better in that benchmark would perform differently in ours. Finally, we believe that methodological concerns with the testing methodology we highlight are independent of specific detector architectures and valid per se.
> >
> > We respectfully disagree with the reviewer's point in W5 regarding the adversarial examples in computer vision. Adversarial evasion literature in computer vision defines adversarial examples as leading the model to deviate from the human interpretation, considered the ground truth. In LLMs, humans are unable to distinguish LLM-generated texts from human-written texts, and have been even for significantly less powerful generative language models that are considered here [5], making automated detection the last and sole line of defense. Similarly, for real-world applications of computer vision, the threat model most of the time requires a physical world pattern to fool the model, which has been proven to be difficult and nonetheless has led to the development of the field of correctness certifiability of machine learning models, ensuring model safety against a threat model with attacker controlling a certain percent of image perturbation budget.
> >
> > Again, we thank the reviewer for their feedback and apologize for the lack of clarity in our original manuscript. We hope that our response fully alleviated the reviewer's concerns. To prevent such confusion, we have revised the 1.4 "Contributions" section to better reflect what we stated in the 5 "Conclusion" section and adjusted our threat model presentation to respond to W3.
> >
> > [1] Dugan, Liam et al. "RAID: A Shared Benchmark for Robust Evaluation of Machine-Generated Text Detectors." Annual Meeting of the Association for Computational Linguistics (2024)
> >
> > [2] Arp, Daniel et al. "Dos and don'ts of machine learning in computer security." USENIX Security 31 (2022)
> >
> > [3] Liang, Weixin et al. "GPT detectors are biased against non-native English writers." Patterns 4 (2023)
> >
> > [4] Metz, Cade "OpenAI Says It Disrupted an Iranian Misinformation Campaign" The New York Times 16.08.2024 (2024)
> >
> > [5] Ippolito, Daphne et al. "Automatic Detection of Generated Text is Easiest when Humans are Fooled." Annual Meeting of the Association for Computational Linguistics (2019)

---

> > > ### Comment · Reviewer_52s7 · 2024-12-03
> > >
> > > I thank the authors for their efforts and response. I have checked the authors' responses both to me and other reviewers, my main concerns have not been properly resolved. I believe there is still much room to improve to meet the criteria of ICLR conference, both in terms of major contribution and its presentation and clarity. Therefore, I will maintain my rating.

---

### Official Review · Reviewer_YjU5 · 2024-11-04

**Soundness:** 3
**Presentation:** 2
**Contribution:** 2
**Rating:** 5
**Confidence:** 4

**Summary:**

This paper examines the effectiveness of existing large language model (LLM) detectors in identifying short news-like posts generated by LLMs in real-world scenarios. The authors find that current zero-shot and purpose-trained detectors are inadequate, particularly when subjected to simple attacks like increased sampling temperature. They highlight the need for domain-specific benchmarking and propose a dynamically extensible benchmark to improve detector evaluation. The study concludes that LLM detectors are not yet ready for real-world application to counter LLM-generated disinformation.

**Strengths:**

1. This paper studies the current LLM-generated text detectors can not work well and conduct benchmark experiments.
2. The attack testing method utilizes high temperature and repetition penalty to reduce the diversity of the produced text, making it more difficult for detection methods that rely on the lack of diversity of AI-generated texts.

**Weaknesses:**

1. The novelty is limited and the results are not surprising.

**Questions:**

1. How does varying the size of LLMs (e.g., 2B-7B parameters) impact the effectiveness of both detection and adversarial evasion?

2. One finding was that detectors trained on specific models generalized better across unseen LLMs but struggled with human-written texts from different distributions. What methods could improve generalization to different domains of human-generated text?

---

> ### Author Response · Authors · 2024-11-28
> **Author response**
>
> We thank the reviewer for their feedback.
>
> While we understand the weakness pointed out by the reviewer, we respectfully disagree. Given the existing literature on the subject of LLM detector benchmarking, we believe that our paper provides three novel conceptual contributions.
>  - There is an inherent trade-off between detector generalization across evasion techniques and generalization across human-written texts. While this trade-off intuitively makes sense, it has neither been stated nor ever benchmarked in the LLM detection literature, which has in the past led to reports of poor performance of detectors on non-native writers [1].
>  - A relatively simple strategy - round-robin - was able to train LLM detectors that generalized well across new LLMs, both conversationally fine-tuned from foundational models in the training set and unrelated families, and across evasion attacks. While previously reported in the context of image-generating GANs, Round-Robin's boost to generalization capabilities in text classification with fine-tuned encoder LLMs has previously been unreported.
>  - Large-scale static benchmarks are not suited for LLM detector evaluation due to the inherently adversarial nature of the setting and high variability of performance metrics across application domains, meaning that even in closely related settings - such as the one we considered and the "news articles" in the RAID benchmark [2], that we highlighted in 4.3 "Prior Benchmark" section. Once again, while not entirely surprising, this has not been previously reported in the context of LLM detectors and suggests large-scale benchmarks that dominated the literature until now might not be useful even for selecting an appropriate detector and suggests a need for a scope in the domain of application and threat model, as we suggest.
>
>
> Given that those conceptual results were dispersed in the introduction, discussion, and conclusion, we adjusted our paper to clarify this field-level contribution in the Abstract and 1.4 "Contributions" section.
>
> Responding to the questions of reviewers:
> Q1: Based on the results in Table 6 (Appendix), we do not observe any clear trends between Gemma-Chat (2B), LLaMA-3-8B-Instruct, and Zephyr-7B-Beta. We believe this suggests that the model size does not impact its detectability or evasion capabilities.
>
> Q2: This is an excellent question. While we believe it is out of scope for this work, the performance of the ELECTRA-RoundRobin detector suggests that applying a similar technique for both human and LLM-generated texts, increasing the diversity of the former might lead to interesting results. Unfortunately, we are unaware of any news text datasets focusing on the diversity and representation of human variability, and we will be thankful for any pointers.
>
> Again, we thank the reviewer for their feedback and apologize for the lack of clarity in our original manuscript. We hope that our response fully alleviated the reviewer's concerns. To prevent such confusion, we have revised the 1.4 "Contributions" section to better reflect what we stated in the 5 "Conclusion" section and adjusted the abstract.
>
> [1] Liang, Weixin et al. "GPT detectors are biased against non-native English writers." Patterns 4 (2023)
>
> [2] Dugan, Liam et al. "RAID: A Shared Benchmark for Robust Evaluation of Machine-Generated Text Detectors." Annual Meeting of the Association for Computational Linguistics (2024).

---

### Meta-Review · Area_Chair_FQvP · 2024-12-17

**Metareview:**

The reviewers unanimously voted to reject this paper, and I tend to agree with the reviewers here.  The reviewers point out that the contributions here are modest, the experiments are narrow, the presentation is lacking, among other criticisms.  The reviewers do believe this paper has the potential to be impactful, so I encourage the authors to improve their work, but I recommend rejection for now.

**Additional Comments On Reviewer Discussion:**

The authors responded to most reviewer feedback during the rebuttal period but were unable to satisfactorily address much of the feedback.

---

### Decision · Program_Chairs · 2025-01-22

Reject